# New Anti-Glycative Lignans from the Defatted Seeds of *Sesamum indicum*

**DOI:** 10.3390/molecules28052255

**Published:** 2023-02-28

**Authors:** Gyeong Han Jeong, Tae Hoon Kim

**Affiliations:** Department of Food Science and Biotechnology, Daegu University, Gyeongsan 38453, Republic of Korea

**Keywords:** *Sesamum indicum* L., defatted sesame seeds, lignan, AGEs formation inhibition, ONOO^−^ scavenging activity

## Abstract

Seven known analogs, along with two previously undescribed lignan derivatives sesamlignans A (**1**) and B (**2**), were isolated from a water-soluble extract of the defatted sesame seeds (*Sesamum indicum* L.) by applying the chromatographic separation method. Structures of compounds **1** and **2** were elucidated based on extensive interpretation of 1D, 2D NMR, and HRFABMS spectroscopic data. The absolute configurations were established by analyzing the optical rotation and circular dichroism (CD) spectrum. Inhibitory effects against the formation of advanced glycation end products (AGEs) and peroxynitrite (ONOO^−^) scavenging assays were performed to evaluate the anti-glycation effects of all isolated compounds. Among the isolated compounds, (**1**) and (**2**) showed potent inhibition towards AGEs formation, with IC_50_ values of 7.5 ± 0.3 and 9.8 ± 0.5 μM, respectively. Furthermore, the new aryltetralin-type lignan **1** exhibited the most potent activity when tested in the in vitro ONOO^−^ scavenging assay.

## 1. Introduction

Among naturally occurring bioactive polyphenolics, lignan is a class of diphenolic secondary metabolites that are distributed in the plant kingdom and biosynthesized by oxidative dimerization of two phenylpropanoid units [1]. This class of natural polyphenols possesses a broad range of structural diversity and biological potencies, including anti-tumor, antioxidant, anti-inflammatory, anti-neurodegenerative, and antiviral activities [2]. Recent interest in novel lignans originated from natural foodstuffs has continuing increase due to their biological benefits correlated with disease prevention and health promotion. Sesame (*Sesamum indicum* L.), one of the most important oilseed crops, is a rich source of lignans such as sesamol, sesamin, sesamolin, and sesaminol [3]. Sesame seeds produce a highly stable oil and provide several food nutritional benefits [4]. Studies have reported good antioxidant activities [5] as well as numerous valuable biological effects [6]. Sesame cake obtained from oil extraction is mainly used as a feed ingredient for domestic animals or is discarded. However, defatted sesame seeds are reported to exert various biological properties, including antioxidant [7], anti-diabetic [8,9], and inhibition of brain edema formation [10], and were found to be rich in lignan and polyphenolic constituents [11,12]. Especially, the major lignan glucoside (sesaminol triglucoside) isolated from defatted sesame cake exhibits potent biological activities such as radical scavenging and cytochrome P450 enzyme inhibition [13,14,15]. Current research interests in the biologically active lignans of *S. indicum* continue because of their inherent ability to prevent diseases and improve health [16,17].

Persistent hyperglycemia is a critical cause associated with the pathogenesis of diabetic complications [18], which can arise from protein kinase C (PKC) isoform activation, increased aldose reductase (AR)-related polyol pathway flux, increased hexosamine pathway flux, and the formation of advanced glycation end products (AGEs) [19]. The formation and accumulation of AGEs are closely implicated in diabetic complication-associated diseases such as arteriosclerosis, kidney disease, neuropathy, osteoporosis, and Alzheimer’s disease [20]. The AGE accumulation reaction is further accelerated with the increased formation of reactive oxygen species (ROS) and free radicals [21]. Thus, suppressing the formation of AGEs and oxidative stress is recognized as an effective therapeutic strategy for diabetic complications in humans.

As part of our continuing search to discover naturally occurring anti-glycation compounds from various natural sources, the ethyl acetate-soluble extract of defatted sesame seeds was found to inhibit activities against the in vitro formation of AGEs and radical scavenging assays. The subsequent bioactivity-guided isolation of this extract led to the isolation of a new aryltertralin-type (**1**) and a tertrahydrofuran-type lignan (**2**) along with seven known compounds (**3**−**9**), using successive column chromatography over Toyopearl HW-40 gel and ODS AQ gel together with semipreparative HPLC. The new structure elucidation, inhibition of AGEs formation, and ONOO^−^ scavenging activity of compounds **1** and **2** were conducted and are presented in this study (Figure 1).

## 2. Results and Discussion

### 2.1. Structure Elucidation of New Compounds and Characterization of Known Compounds

The new structure compound **1** was isolated as a yellow amorphous powder, [α]^20^_D_ +21.6° (*c* 0.1, MeOH). A pseudomolecular ion peak at *m*/*z* 360.1206 [M]^+^, observed in the HRFABMS spectrum of **1** in conjunction with ^13^C NMR spectroscopic data, corresponded to the molecular formula C_19_H_20_O_7_. The ^1^H NMR spectrum of **1** displayed four aromatic proton signals at *δ*_H_ 6.52 (1H, s, H-2), 6.40 (1H, s, H-3′), 6.29 (1H, s, H-6′), and 6.18 (1H, s, H-5), revealing the presence of two 1,3,4,6-tetrasubstituted aromatic rings. Signals for a methylenedioxy group at *δ*_H_ 5.78 (2H, s, H-10′), two oxygenated methylene proton signals at *δ*_H_ 3.69 (1H, m, H-9′), 3.64−3.62 (2H, m, H-9), and 3.42 (1H, dd, *J* = 11.4, 4.8 Hz, H-9′), one methylene at *δ*_H_ 2.68 (2H, d, *J* = 7.8 Hz, H-7), and three methines at *δ*_H_ 4.17 (1H, d, *J* = 10.2 Hz, H-7′), 1.94 (1H, m, H-8), and 1.73 (1H, m, H-8′) were also obtained on the ^1^H NMR spectrum (Table 1). The ^13^C NMR and HSQC spectra of **1** showed 19 carbon signals: 12 aromatic carbon signals at *δ*_C_ 151.2 (C-2′), 147.5 (C-5′), 144.3 (C-4), 144.2 (C-3), 142.4 (C-4′), 132.1 (C-1), 129.4 (C-6), 125.3 (C-1′), 117.2 (C-5), 115.7 (C-2), 109.9 (C-3′), and 98.4 (C-6′), one methylenedioxy group at *δ*_C_ 102.0 (C-10′), two oxygenated methylenes at *δ*_C_ 66.2 (C-9) and 63.3 (C-9′), one methylene at *δ*_C_ 33.4 (C-7), and three methine carbons at *δ*_C_ 47.8 (C-8′), 40.9 (C-7′), and 40.8 (C-8) (Table 1). Analogous resonances consistent with the presence of these functionalities were displayed in the ^13^C NMR data of **1** (Table 1) [22,23]. The ^1^H-^1^H COSY correlations of H-7/H-8/H-9, H-7′/H-8′/H-9′, and H-8/H-8′, and the HMBC correlations of H-8 to C-1,-7,-7′ and H-7′ to C-1′,-2,-2′,-8,-8′,-9′, indicate that **1** is an aryltetralin-type lignan [24]. The HMBC correlations from the methylenedioxy moiety (H-10′) indicate that this is located at C-4′ (*δ*_C_ 142.4) and -5′ (*δ*_C_ 147.5) (Figure 2A) [24].

The relative configuration of **1**, established on the basis of analyzing the NOESY correlations observed of H-6′/H-8′, H-7′/H-8, and H-7′/H-9′ along with the large-coupling constants (*J*_7′, 8′_ = 10.2 Hz), confirmed the *trans*-*trans* arrangement (Figure 2B) [25,26]. Compared to previous reports, the circular dichroism (CD) spectrum of **1** showed positive Cotton effects at 238 (Δε + 4.7) and 282 (Δε + 2.4) nm and a negative Cotton effect at 301 (Δε −8.7) nm (Appendix A) [27]. Consequently, the absolute configuration of **1** was determined to be a 7′*S*, 8*R*, 8′*R*-configuration. Compound **1** was identified as a new naturally occurring arytetralin-type lignan, and this compound was assigned the trivial name sesamlignan A (Appendix A).

Compound **2** was isolated as a brown amorphous optically active powder ([α]^20^_D_ +58.9°) and showed a pseudomolecular ion at *m*/*z* 344.1262 [M]^+^, corresponding to the molecular formula C_19_H_20_O_6_ in the HRFABMS. The ^1^H NMR data of **2** exhibited resonances for two sets of ABX-type aromatic rings at *δ*_H_ 6.82 (1H, d, *J* = 2.4 Hz, H-2′), 6.77 (1H, dd, *J* = 7.8, 2.4 Hz, H-6′), 6.74 (1H, d, *J* = 7.8 Hz, H-6′), 6.67 (1H, d, *J* = 8.4 Hz, H-5), 6.62 (1H, d, *J* = 2.4 Hz, H-2), and 6.51 (1H, dd, *J* = 8.4, 2.4 Hz, H-6), and one methylenedioxy group at *δ*_H_ 5.90 (2H, s, H-10′). The spectrum also included signals attributable to three methine protons at *δ*_H_ 4.76 (1H, d, *J* = 6.0 Hz, H-7′), 2.68 (1H, m, H-8), and 2.29 (1H, m, H-8′), one methylene proton at *δ*_H_ 2.81 (1H, dd, *J* = 13.8 Hz, 5.4, H-7) and 2.42 (1H, dd, *J* = 13.8, 10.2 Hz, H-7), one oxygenated methylene protons at 3.97 (1H, dd, *J* = 8.4, 6.6 Hz, H-9), 3.69 (1H, dd, *J* = 8.4, 6.6 Hz, H-9), and one hydroxymethyl group at *δ*_H_ 3.81 (1H, dd, *J* = 10.8, 7.2 Hz, H-9′) and 3.61 (1H, dd, *J* = 10.8, 7.2 Hz, H-9′) (Table 1). These NMR data observations indicate that **2** contains a three-ring system: two aromatic rings and one tetrahydrofuran moiety. The above data, together with the observation of ^13^C NMR and HSQC spectra, implied that **2** is a tetrahydrofuran-type lignan [28]. The connective positions of each methylenedioxyphenyl and a dihydroxyphenyl group by the HMBC technique, which demonstrated a three-bond correlation between the oxygenated methine proton (H-7′) to C-1′ (*δ*_C_ 138.6), -2′ (*δ*_C_ 107.2), and -6′ (*δ*_C_ 121.0), and methylene proton signals (H-7) to C-1 (*δ*_C_ 133.5), -2 (*δ*_C_ 108.6), -6 (*δ*_C_ 120.3), -8 (*δ*_C_ 43.7), and -9 (*δ*_C_ 73.7) positions, respectively (Figure 2A). The linkage point of hydroxymethyl residue (H-9′) on the furan moiety (C-7′, -8, -8′) in **2** was determined unambiguously from key HMBC correlations in Figure 2A. These spectroscopic features are comparable to those reported for acuminatin [29], previously isolated from the aerial parts of *Helichrysum acuminatum*, except for the signals of a methoxyl group at C-3 present in **2**.

Moreover, the proposed relative stereochemistry of **2** in the tetrahydrofuran moiety was confirmed by the spatial correlations between H-7′, H-7, and H-9′, H-8 and H-8′, and H-8 and H-9 as observed in the NOESY spectrum (Figure 2B) [30,31]. The absolute configuration of the tetrahydrofuran moiety in **2** was determined as a 7′*S*, 8*R*, 8′*R*-configuration based on the positive specific optical rotation value {[α]^20^_D_ +58.9° (*c* 0.1, MeOH)} as well as negative Cotton effects at 229 (Δε −0.3) and 288 (Δε −0.7) nm in the CD spectral comparison using authentic analogs (Appendix A) [32]. Although the planar structure of **2** has previously been reported as an intermediate of sesamin metabolites identified in rat urine by GC-MS data [33], this is the first report of isolation and determination of the absolute structure using spectroscopic interpretation (Appendix A).

Based on the spectroscopic analysis and comparison of the data with literature values, the previously reported compounds **3**−**9** were identified as (+)-sesaminol 2′-*O*-glucopyranosyl(1→2)-*O*-[glucopyranosyl(1→6)]-*O*-glucopyranoside (**3**) [11,34], (+)-sesaminol 2′-*O*-glucopyranosyl(1→2)-*O*-glucopyranoside (**4**) [13,35], (+)-sesaminol (**5**) [36], (+)-epipinoresinol 4′-*O*-glucopyranoside (**6**) [37], (+)-epipinoresinol (**7**) [38], apocynin (**8**) [39], and vanillic acid (**9**) [10] (Appendix A). The methyl-substituted vanillic acid, apocynin (**8**), was first isolated from *Sesamum indicum* (Figure 1).

### 2.2. Inhibition of Formation of AGEs and ONOO^−^ Scavenging Effects

All pure isolated compounds **1**−**9** were evaluated for their capacity to inhibit the formation of AGEs using aminoguanidine as the positive control (Table 2). Compared to the positive control aminoguanidine (IC_50_: 995.3 ± 3.6 μM), the most potent inhibitory effects against AGEs formation were exhibited by the new lignans sesamelignans A (**1**) and B (**2**) with IC_50_ values of 7.5 ± 0.3 and 9.8 ± 0.5 μM, respectively. The IC_50_ values of the furfuran-type lignans **3**−**7** were obtained in the range 17.8 to 65.8 μM for AGEs formation ability. The simple phenolic compounds **8** and **9** were considerably less effective compared to other lignan derivatives. In addition, we further evaluated the anti-oxidant effects of the isolated compounds using the previously reported ONOO^−^ scavenging assay [40]. The novel aryltetralin lignan **1** showed maximum scavenging activity against the ONOO^−^ scavenging assay (IC_50_: 8.1 ± 0.5 μM) compared to the positive control _L_-penicillamine (IC_50_: 15.0 ± 1.0 μM). Vanillic acid (**9**) has previously been described as a powerful ONOO^−^ scavenging substance isolated from *Panax ginseng*, and our results are in agreement with those findings [41]. Although various lignan analogs from natural products have been reported as anti-glycation inhibitors [42], the current study is the first to validate the new aryltetralin-type lignan **1** with potent inhibitory effects of AGEs formation and ONOO^−^ scavenging activity. Taken together, our results indicate the potential to develop sesamelignan A (**1**) as a therapeutic for diabetic complications and related diseases.

## 3. Materials and Methods

### 3.1. General Experimental Procedures

The ultraviolet (UV) spectrum was measured on a T-60 spectrophotometer (PG Instrument, Leicestershire, UK), and the circular dichroism (CD) spectrum was run on a JASCO J-1500 spectrometer (JASCO, Tokyo, Japan). ^1^H-, ^13^C-NMR, ^1^H-^1^H COSY, HSQC, HMBC, and NOESY spectra were measured on a Varian VNS-600 MHz spectrometer (Varian, Palo Alto, CA, USA) equipment using CD_3_OD (*δ*_H_ 3.35, *δ*_C_ 49.0) as the solvent and tetramethylsilane (TMS) as the internal standard. Fast atom bombardment mass spectrometer (FABMS) was recorded on a JMS-700 GC-HRMS spectrometer (JEOL, Tokyo, Japan), and optical rotation was obtained using a JASCO P-2000 polarimeter. Toyopearl HW-40C gel (Tosho Co. Tokyo, Japan) and ODS gel (ODS AQ 120-50S, YMC Co., Kyoto, Japan) were used for column chromatography.

### 3.2. Plant Material and Preparation

Sesame seeds (*Sesamum indicum* L.) were collected in June 2017 from Yecheon-gun, Republic of Korea, and identified by Prof. Tae Hoon Kim. A voucher specimen was deposited at the Natural Products Chemistry Laboratory of Daegu University. The dried sesame seeds (20 kg) were roasted in an electric frying pan (D-1692, Dongkwang oil machine Co., Seoul, Korea) at 300 °C for 12 min. Oil was extracted from the roasted sesame seeds using an electric oil squeezer (D-1880, Dongkwang oil machine Co., Seoul, Korea), and the remaining sesame byproducts were used in the experiment.

### 3.3. Extraction and Isolation

Defatted sesame seeds (8.0 kg) were powdered and extracted with distilled water (40 L) at 70 °C for 3 h, after which the extract solution was concentrated in vacuo to yield the solid extract (726.0 g). The dried extract (720.0 g) was suspended in 10% MeOH in H_2_O (1 L) and partitioned sequentially using organic solvents to yield *n*-hexane—(2.3 g), EtOAc—(32.2 g), *n*-BuOH—(66.9 g), and H_2_O—(425.1 g) soluble fractions. The EtOAc−soluble fraction was found to be active in the AGEs formation inhibition assay, with an IC_50_ value of 154.8 ± 2.4 μg/mL (Appendix A). One portion of the EtOAc−fraction (23.5 g) was chromatographed over a Toyopearl HW-40 column (4 cm i.d. × 40 cm, coarse grade) eluted with gradient systems of H_2_O-MeOH increasing polarity (0% to 100%, followed by 70% acetone) to yield eleven sub-fractions (SE01-SE11). Fraction SE03 (480.1 mg) was subjected to ODS gel column chromatography (1 cm i.d. × 40 cm, particle size 50 μm) with a MeOH/H_2_O system, resulting in the isolation of compounds **3** (55.3 mg, *t*_R_ 20.9 min), **4** (54.1 mg, *t*_R_ 23.2 min), and **6** (37.4 mg, *t*_R_ 19.0 min). Similar fractionation of SE04 (306.6 mg) on ODS gel chromatography (1 cm i.d. × 42 cm) yielded the pure compounds **8** (13.8 mg, *t*_R_ 16.4 min) and **9** (15.5 mg, *t*_R_ 10.9 min). Finally, the sub-fraction SE08 (155.9 mg) was subjected to ODS gel column chromatography (1 cm i.d. × 42 cm) with aqueous MeOH to give the pure compounds **1** (4.7 mg, *t*_R_ 16.8 min), **2** (2.6 mg, *t*_R_ 22.6 min), **5** (3.4 mg, *t*_R_ 31.0 min), and **7** (3.9 mg, *t*_R_ 25.0 min). HPLC (Shimadzu, Tokyo, Japan) analysis was performed using the YMC-Pack ODS A-302 column (4.6 mm i.d. × 150 mm, particle size 5 μm; YMC Co., Kyoto, Japan) and mobile phase comprising 0.1% HCOOH in H_2_O (Solvent A) and MeCN (Solvent B). A gradient system was performed with a linear gradient of 5% to 100% solvent B for 35 min with the flow rate set at 1.0 mL/min.

Sesamlignan A (**1**): Yellow amorphous powder. [α]^20^_D_ +21.6° (*c* 0.1, MeOH). UV λ_max_ MeOH (log ε): 205 (3.54), 235 (sh), 295 (1.19) nm. CD (MeOH) Δε (nm): 211 (+21.3), 238 (+4.7), 282 (+2.4), 301 (−8.7) nm. ^1^H- and ^13^C-NMR: see Table 1. FABMS *m*/*z* 360 [M]^+^. HRFABMS *m*/*z* 360.1203 [M]^+^ (calc. for C_19_H_20_O_7_, 360.1209) (Appendix A).

Sesamlignan B (**2**): Brown amorphous powder. [α]^20^_D_ +58.9° (*c* 0.1, MeOH). UV λ_max_ MeOH (log ε): 204 (3.75), 234 (sh), 285 (1.10) nm. CD (MeOH) Δε (nm): 210 (−4.3), 229 (−0.3), 288 (−0.7) nm. ^1^H- and ^13^C-NMR: see Table 1. FABMS *m*/*z* 344 [M]^+^. HRFABMS *m*/*z* 344.1262 [M]^+^ (calc. for C_19_H_20_O_6_, 344.1260) (Appendix A).

(+)-Sesaminol 2′-*O*-glucopyranosyl(1→2)-*O*-[glucopyranosyl(1→6)]-*O*-glucopyrano side (**3**): Yellow amorphous powder. [α]^20^_D_ −47.0° (*c* 0.1, MeOH). ^1^H-NMR (CD_3_OD, 600 MHz): *δ* 6.91 (1H, s, H-3′), 6.85 (1H, d, *J* = 1.2 Hz, H-2″), 6.83 (1H, s, H-6′), 6.81 (1H, dd, *J* = 7.8, 1.2 Hz, H-6″), 6.76 (1H, d, *J* = 7.8 Hz, H-5″), 5.91 (2H, s, H-7″), 5.89 (2H, s, H-7′), 5.20 (1H, d, *J* = 4.8 Hz, H-2), 5.02 (1H, d, *J* = 7.2 Hz, H-1″′), 4.87 (1H, d, *J* = 7.2 Hz, H-1″″), 4.69 (1H, d, *J* = 4.8 Hz, H-6), 4.34 (1H, d, *J* = 7.2 Hz, H-1″″′), 4.24 (1H, dd, *J* = 9.0, 6.6 Hz, H-4_eq_), 4.21 (1H, d, *J* = 10.8 Hz, H-8_ax_), 4.19 (1H, d, *J* = 10.8 Hz, H-8_eq_), 4.13 (1H, dd, *J* = 10.8, 1.2 Hz, H-6″′), 3.81 (1H, overlap, H-4_ax_), 3.86–3.19 (17H, overlap, glucose), 3.00 (1H, m, H-5), 2.90 (1H, m, H-1); ^13^C-NMR (CD_3_OD, 150 MHz): *δ* 149.9 (C-2′), 149.3 (C-5′), 148.5 (C-4″), 148.4 (C-3″), 144.0 (C-4′), 136.5 (C-1″), 125.5 (C-1′), 120.7 (C-6″), 109.0 (C-5″), 107.6 (C-2″), 105.9 (C-6′), 104.9 (C-1″″′), 104.5 (C-1″″), 102.6 (C-7′), 102.4 (C-7″), 101.2 (C-1″′), 99.3 (C-3′), 86.5 (C-6), 82.8 (C-2), 81.8 (C-2″′), 78.4 (C-3″′), 78.1 (C-4″″), 78.0 (C-3″′), 77.9 (C-3″″′), 77.8 (C-5″″), 76.9 (C-5″′), 76.0 (C-2″″′), 75.1 (C-2″″), 73.8 (C-8), 72.8 (C-4), 71.6 (C-4″″′), 71.2 (C-5″″′), 71.1 (C-4″′), 70.3 (C-6″′), 62.8 (C-6″″′), 62.2 (C-6″″), 55.8 (C-1), 55.6 (C-5). FABMS *m*/*z* 856 [M]^+^ (Appendix A).

(+)-Sesaminol 2′-*O*-glucopyranosyl(1→2)-*O*-glucopyranoside (**4**): Yellow amorphous powder. [α]^20^_D_ −26.7° (*c* 0.1, MeOH). ^1^H-NMR (CD_3_OD, 600 MHz): *δ* 6.91 (1H, s, H-3′), 6.85 (1H, d, *J* = 1.8 Hz, H-2″), 6.81 (1H, s, H-6′), 6.80 (1H, dd, *J* = 7.8, 1.8 Hz, H-6″), 6.75 (1H, d, *J* = 7.8 Hz, H-5″), 5.90 (2H, s, H-7″), 5.89 (2H, s, H-7′), 5.18 (1H, d, *J* = 4.8 Hz, H-2), 4.84 (1H, d, *J* = 7.8 Hz, H-1″′), 4.62 (1H, d, *J* = 4.8 Hz, H-6), 4.34 (1H, d, *J* = 7.8 Hz, H-1″″), 4.27 (1H, dd, *J* = 9.6, 7.8 Hz, H-8_eq_), 4.17 (1H, dd, *J* = 9.6, 6.0 Hz, H-4_eq_), 4.13 (1H, m, H-6″′), 4.06 (1H, dd, *J* = 9.6, 4.8 Hz, H-8_ax_), 3.84 (1H, dd, *J* = 9.6, 4.8 Hz, H-4_ax_), 3.86–3.21 (11H, overlap, glucose), 3.00 (1H, m, H-5), 2.95 (1H, m, H-1); ^13^C-NMR (CD_3_OD, 150 MHz): *δ* 150.4 (C-2′), 149.3 (C-5′), 148.6 (C-4″), 148.5 (C-3″), 144.1 (C-4′), 136.4 (C-1″), 125.5 (C-1′), 120.7 (C-6″), 109.0 (C-5″), 107.6 (C-2″), 106.0 (C-6′), 104.8 (C-1″″), 103.2 (C-7′), 102.6 (C-7″), 102.4 (C-1″′), 99.9 (C-3′), 86.9 (C-6), 82.9 (C-2), 78.1 (C-5″′), 78.0 (C-5″″), 77.1 (C-3″′), 75.0 (C-4″′), 74.9 (C-2″″), 74.2 (C-2″′), 72.4 (C-3″″), 71.6 (C-4″″), 71.3 (C-4), 71.2 (C-8), 70.8 (C-6″′), 62.8 (C-6″″), 55.6 (C-1), 55.2 (C-5). FABMS *m*/*z* 694 [M]^+^ (Appendix A).

(+)-Sesaminol (**5**): White amorphous powder. [α]^20^_D_ +73.6° (*c* 0.1, MeOH). ^1^H-NMR (CD_3_OD, 600 MHz): *δ* 6.86 (1H, d, *J* = 1.2 Hz, H-2″), 6.82 (1H, dd, *J* = 7.2, 1.2 Hz, H-6″), 6.76 (1H, d, *J* = 7.2 Hz, H-5″), 5.91 (2H, s, H-7″), 6.75 (1H, s, H-3′), 6.35 (1H, s, H-6′), 5.81 (2H, s, H-7′), 4.98 (1H, d, *J* = 4.2 Hz, H-2), 4.67 (1H, d, *J* = 4.2 Hz, H-6), 4.21 (1H, dd, *J* = 9.6, 7.8 Hz, H-8_eq_), 4.24 (1H, dd, *J* = 9.6, 6.0 Hz, H-4_eq_), 4.01 (1H, dd, *J* = 9.6, 4.2 Hz, H-8_ax_), 3.85 (1H, dd, *J* = 9.6, 4.2 Hz, H-4_ax_), 3.00 (1H, m, H-5), 2.95 (1H, m, H-1); ^13^C-NMR (CD_3_OD, 150 MHz): *δ* 151.2 (C-2′), 148.2 (C-5′), 147.7 (C-4″), 146.4 (C-3″), 145.0 (C-4′), 135.4 (C-1″), 126.1 (C-1′), 122.1 (C-6″), 109.2 (C-5″), 107.8 (C-2″), 106.5 (C-6′), 102.9 (C-7′), 101.4 (C-7″), 99.8 (C-3′), 85.8 (C-6), 81.0 (C-2), 71.6 (C-4), 70.5 (C-8), 53.2 (C-1), 51.0 (C-5). FABMS *m*/*z* 370 [M]^+^ (Appendix A).

(+)-Epipinoresinol 4′-*O*-glucopyranoside (**6**): Brown amorphous powder. [α]^20^_D_ +37.7° (*c* 0.1, MeOH). ^1^H-NMR (CD_3_OD, 600 MHz): *δ* 7.13 (1H, d, *J* = 8.4 Hz, H-5″), 7.02 (1H, d, *J* = 1.8 Hz, H-2″), 6.63 (1H, d, *J* = 1.8 Hz, H-2′), 6.91 (1H, dd, *J* = 8.4, 1.8 Hz, H-6″), 6.80 (1H, dd, *J* = 7.8, 1.8 Hz, H-6′), 6.75 (1H, d, *J* = 7.8 Hz, H-5′), 4.82 (1H, d, *J* = 7.8 Hz, H-1″′), 4.74 (1H, d, *J* = 4.8 Hz, H-2), 4.69 (1H, d, *J* = 7.2 Hz, H-6), 4.23 (1H, overlap, H-8_eq_), 4.22 (1H, overlap, H-4_eq_), 3.86 (3H, s, OCH_3_-3″), 3.84 (3H, s, OCH_3_-3′), 3.82 (1H, m, H-8_ax_), 3.80 (1H, m, H-4_ax_), 3.69 (1H, dd, *J* = 11.4, 4.2 Hz, H-6″′), 3.68 (1H, m, H-6″′), 3.46 (1H, m, H-2″′), 3.45 (1H, m, H-4″′), 3.30 (1H, m, H-3″′), 3.29 (1H, m, H-5″′), 3.12 (1H, m, H-5), 3.10 (1H, m, H-1); ^13^C-NMR (CD_3_OD, 150 MHz): *δ* 151.1 (C-3″), 149.1 (C-3′), 147.5 (C-4″), 147.3 (C-4′), 137.5 (C-1″), 133.8 (C-1′), 120.1 (C-6′), 119.8 (C-6″), 118.1 (C-5″), 116.1 (C-5′), 111.7 (C-2′), 111.0 (C-2″), 102.9 (C-1″′), 87.5 (C-6), 87.1 (C-2), 78.2 (C-5″′), 77.9 (C-3″′), 74.9 (C-2″′), 73.1 (C-4), 72.7 (C-8), 71.3 (C-4″′), 62.5 (C-6″′), 56.8 (OCH_3_-3′), 56.4 (OCH_3_-3″), 55.5 (C-1), 55.4 (C-5). FABMS *m*/*z* 520 [M]^+^ (Appendix A).

(+)-Epipinoresinol (**7**): Brown amorphous powder. [α]^20^_D_ +84.5° (*c* 0.1, MeOH). ^1^H-NMR (CD_3_OD, 600 MHz): *δ* 6.94 (1H, d, *J* = 1.8 Hz, H-2″), 6.96 (1H, d, *J* = 1.8 Hz, H-2′), 6.81 (1H, dd, *J* = 8.4, 1.8 Hz, H-6″), 6.80 (1H, d, *J* = 8.4 Hz, H-5′), 6.77 (1H, d, *J* = 8.4 Hz, H-5″), 6.76 (1H, dd, *J* = 8.4, 1.8 Hz, H-6′), 4.85 (1H, d, *J* = 4.8 Hz, H-2), 4.41 (1H, d, *J* = 7.2 Hz, H-6), 4.09 (1H, d, *J* = 9.6 Hz, H-8_eq_), 3.83 (1H, d, *J* = 9.6 Hz, H-8_ax_), 3.86 (3H, s, OCH_3_-3″), 3.85 (3H, s, OCH_3_-3′), 3.78 (1H, t, *J* = 9.0 Hz, H-4_eq_), 3.38 (1H, m, H-4_ax_), 3.28 (1H, m, H-5), 2.93 (1H, m, H-1); ^13^C-NMR (CD_3_OD, 150 MHz): *δ* 149.1 (C-3″), 148.8 (C-3′), 147.4 (C-4″), 146.6 (C-4′), 133.9 (C-1″), 131.3 (C-1′), 120.1 (C-6′), 119.4 (C-6″), 116.1 (C-5″), 116.0 (C-5′), 110.9 (C-2′), 111.6 (C-2″), 89.5 (C-6), 83.5 (C-2), 71.9 (C-4), 70.6 (C-8), 56.4 (OCH_3_-3′), 55.6 (OCH_3_-3″), 51.2 (C-1), 59.8 (C-5). FABMS *m*/*z* 358 [M]^+^ (Appendix A).

Apocynin (**8**): White amorphous powder. ^1^H-NMR (CD_3_OD, 600 MHz): *δ* 7.56 (1H, dd, *J* = 8.4, 1.8 Hz, H-6), 7.55 (1H, d, *J* = 1.8 Hz, H-2), 6.85 (1H, d, *J* = 8.4 Hz, H-5), 3.86 (3H, s, OCH_3_-3), 2.53 (3H, s, CH_3_-8); ^13^C-NMR (CD_3_OD, 150 MHz): *δ* 199.4 (C-7), 153.4 (C-4), 149.0 (C-3), 130.6 (C-1), 125.2 (C-6), 115.8 (C-5), 112.0 (C-2), 56.3 (OCH_3_-3), 26.2 (CH_3_-8). FABMS *m*/*z* 166 [M]^+^ (Appendix A).

Vanillic acid (**9**): White amorphous powder. ^1^H-NMR (CD_3_OD, 600 MHz): *δ* 7.55 (1H, d, *J* = 1.8 Hz, H-2), 7.55 (1H, dd, *J* = 8.4, 1.8 Hz, H-6), 6.83 (1H, d, *J* = 8.4 Hz, H-5), 3.87 (3H, s, OCH_3_-3). FABMS *m*/*z* 168 [M]^+^ (Appendix A).

### 3.4. Evaluation of AGEs Formation Inhibitory Effects

Using a previously reported method [43] with minor modification, we evaluated the AGEs formation inhibitory potential of the isolated compounds. Briefly, the reaction mixture was prepared by adding 10 mg/mL BSA in 50 mM phosphate buffer (pH 7.4) containing 0.02% sodium azide to a sugar solution (200 mM _D_-fructose and 200 mM _D_-glucose). The reaction mixture (800 μL) was then combined with various concentrations of either the test compounds (200 μL) or the positive control (aminoguanidine) dissolved in 5% DMSO. After incubation at 37 °C for 7 days, the fluorescent reaction products were determined using an ELISA reader (Infinite F200; Tecan Austria GmBH, Grodig, Austria), with excitation and emission maxima at 350 and 450 nm, respectively. The concentration required for 50% inhibition (IC_50_ value) of the fluorescence intensity was determined by linear regression analysis. All measurements were obtained in triplicate.

### 3.5. Evaluation of ONOO^−^ Scavenging Activities

The ONOO^−^ scavenging ability was evaluated by observing the extremely fluorescent dihydrorhodamine 123 (DHR 123) that is rapidly generated from non-fluorescent DHR 123 in the presence of ONOO^−^ [40]. The dihydrorhodamine buffer (pH 7.4) comprises 50 mM sodium phosphate monobasic, 50 mM sodium phosphate dibasic, 90 mM sodium chloride, 5 mM potassium chloride, and 100 μM DTPA, and the final DHR 123 concentration used was 5.0 μM. The test sample was dissolved in 5% DMSO. The final fluorescent intensities of the treated samples were observed 5 min after treatment with and without the addition of authentic ONOO^−^ (10 μM) dissolved in 0.3 N NaOH. The fluorescence intensity of the oxidized DHR 123 was estimated with a fluorescence ELISA reader at emission and excitation wavelengths of 530 and 480 nm, respectively. Results of the ONOO^−^ scavenging effect were evaluated as the final fluorescence intensity minus the background fluorescence, determined by the detection of DHR 123 oxidation. The 50% inhibition (IC_50_ value) was measured by linear regression analysis of the scavenging activity under the above assay conditions. _L_-Penicillamine was used as a positive control. All measurements were obtained in triplicate.

### 3.6. Statistical Analysis

Data for the in vitro analyses of AGEs formation and ONOO^−^ scavenging activity were analyzed using the Proc GLM procedure of SAS software (version 9.3, SAS Institute Inc., Cary, NC, USA). The results are reported as the least square mean values and standard deviation. Statistical significance was considered at *p* < 0.05.

## 4. Conclusions

This paper reported two previously undescribed lignans (**1** and **2**) along with seven known compounds (**3**−**9**) isolated from the defatted sesame cake. The new chemical structures of **1** and **2** were characterized by detailed NMR, MS, and CD spectra data analysis. All compounds were evaluated for their inhibitory potential against AGEs formation and ONOO^−^ scavenging properties. The unusual aryltetralin-type (**1**) and tertrahydrofuran-type lignan (**2**) showed the most potent inhibitory effects of AGEs formation compared to the positive control. In addition, the newly discovered sesamlignan A (**1**) exhibited a maximum potency for ONOO^−^ scavenging capacity. Thus, we propose that sesamlignans A and B have the potential to be developed as therapeutic agents for treating diabetic complications and related diseases.

## Figures and Tables

**Figure 1 molecules-28-02255-f001:**
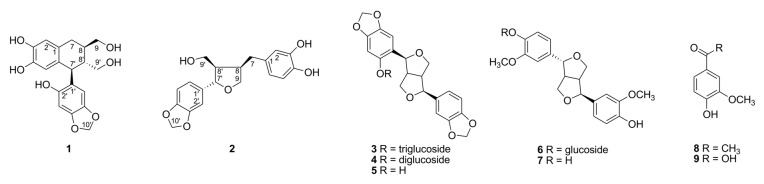
Chemical structures of compounds **1**–**9** isolated from defatted sesame seeds.

**Figure 2 molecules-28-02255-f002:**
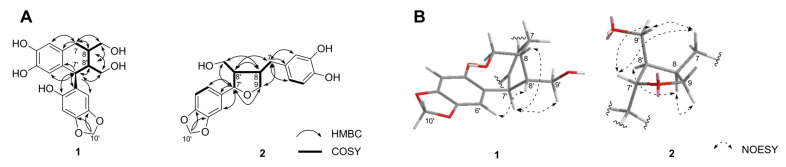
^1^H-^1^H COSY (**A**), key HMBC (**A**), and NOESY (**B**) correlations of **1** and **2**.

**Table 1 molecules-28-02255-t001:** ^1^H and ^13^C NMR data of compounds **1** and **2** in CD_3_OD ^1^.

	1		2	
Positions	*δ*_H_ (*J* in Hz) ^2^	*δ*_C_, Type	*δ*_H_ (*J* in Hz)	*δ*_C_, Type
1	―	132.1, C	―	133.5, C
2	6.52 (s)	115.7, CH	6.62 (d, 2.4)	108.8, CH
3	―	144.2, C	―	144.6, C
4	―	144.3, C	―	146.3, C
5	6.18 (s)	117.2, CH	6.67 (d, 8.4)	116.4, CH
6	―	129.4, C	6.51 (dd, 8.4, 2.4)	120.3, CH
7	2.69 (d, 7.8)	33.4, CH_2_	2.81 (dd, 13.8, 5.4) 2.42 (dd, 13.8, 10.2)	33.3, CH_2_
8	1.94 (m)	40.8, CH	2.68 (m)	43.7, CH
9	3.64 (m) 3.62 (m)	66.2, CH_2_	3.97 (dd, 8.4, 6.6) 3.69 (dd, 8.4, 6.6)	73.7, CH_2_
1′	―	125.3, C	―	138.6, C
2′	―	151.2, C	6.82 (d, 2.4)	107.2, CH
3′	6.40 (s)	98.4, CH	―	148.4, C
4′	―	142.4, C	―	149.3, C
5′	―	147.5, C	6.74 (d, 7.8)	116.7, CH
6′	6.29 (s)	109.9, CH	6.77 (dd, 7.8, 2.4)	121.0, CH
7′	4.17 (d, 10.2)	40.9, CH	4.76 (d, 6.6)	84.0, CH
8′	1.73 (m)	47.8, CH	2.29 (m)	54.1, CH
9′	3.69 (m) 3.42 (dd, 11.4, 4.8)	63.3, CH_2_	3.81 (dd, 10.8, 7.2) 3.61 (dd, 10.8, 7.2)	60.4, CH_2_
10′	5.78 (s)	102.0, CH_2_	5.90 (s)	102.3, CH_2_

^1^ Assignments of chemical shifts are based on the analysis of HSQC and HMBC spectra. ^2^
*J* values (Hz) are given in parentheses.

**Table 2 molecules-28-02255-t002:** Effects on the inhibition of AGEs formation and ONOO^−^ scavenging activities of the isolated compounds **1**–**9**.

Compounds	IC_50_ Value (μM) ^1^	
	Inhibition of AGEs Formation ^2^	ONOO^−^ Scavenging Activity ^2^
**1**	7.5 ± 0.3 ^e^	8.1 ± 0.5 ^e^
**2**	9.8 ± 0.5 ^d^	20.8 ± 0.6 ^c^
**3**	17.8 ± 0.9 ^c^	35.1 ± 0.8 ^c^
**4**	41.7 ± 1.3 ^b^	52.2 ± 1.5 ^b^
**5**	65.8 ± 2.9 ^b^	82.1 ± 2.2 ^a^
**6**	29.0 ± 1.7 ^c^	51.9 ± 1.7 ^b^
**7**	52.4 ± 2.7 ^b^	75.9 ± 2.0 ^a^
**8**	>300 ^a^	59.2 ± 1.2 ^b^
**9**	>300 ^a^	15.5 ± 0.5 ^d^
Aminoguanidine ^3^	995.3 ± 3.6 ^a^	―
_L_-Penicillamine ^3^	―	15.0 ± 1.0 ^d^

^1^ All compounds were examined in triplicate experiments. ^2^ Different letters (a–e) within the same column indicate significant differences (*p* < 0.05). ^3^ Aminoguanidine and L-penicillamine were used as positive controls.

## Data Availability

The data presented in this study are available on request from the corresponding author.

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
