# Peer review of "New Anti-Glycative Lignans from the Defatted Seeds of Sesamum indicum"

_molecules, 2023, doi:10.3390/molecules28052255_

Round 1

Reviewer 1 Report

Dear Authors,

The manuscript presents a theme of significant scientific relevance.

Author Response

Dear Authors, The manuscript presents a theme of significant scientific relevance.

: Thank you for your kind and valuable comments to our manuscript. 

Reviewer 2 Report

The researchers studied the composition of the defatted sesame seeds, resulting in the discovery of two new lignans named sesamlignans A (1) and B (2) together with seven known analogs. Moreover, sesamlignans A (1) and B (2) exhibited potent inhibitory effects against AGEs formation and ONOO scavenging activities, indicating these two compounds have the potential to be developed as therapeutic agents for treating diabetic complications and related diseases. Based on these intriguing findings, this manuscript will gain interests from readers. However, revisions are necessary before this manuscript is accepted.

1. P2L67, P6L210: As shown in Figure S7, the observed m/z was 360.1206 not 360.1203.

2. P2L73: According to Table 1, the chemical shift δH 3.42 ppm was attributed to H-9' not H-9.

3. P2L84: The HMBC correlation of H-7' to C-2 was questionable. In my opinion, the correct one was H-7' to C-5. However, Figure S5 was too small for me to analyze it. Please check it.

4. The citations [22] and [23] were missing before [24] in the text.

5. P2L85: The carbons C-4' and C-5' were missing when referred to ‘The HMBC correlations from the methylenedioxy moiety (H-10') indicate...’.

6. P3: The relative configuration of C-8 was not determined, which should be done before the elucidation of its absolute configuration. Please check it.

7. For compound 2, the interpretation of the methylene H2-9 in its 1H NMR spectrum was missing. And describe the key HMBC correlations in detail, not briefly mentioned as ‘Figure 2A’. The relative configuration of C-7' was not clearly determined.

8. The citations [33] and [34] were missing before [35] in the text.

Author Response

  1. P2L67, P6L210: As shown in Figure S7, the observed m/z was 360.1206 not 360.1203.

→ We corrected mentioned mass data (m/z 360.1203 → m/z 360.1206) and showed as red-color.

  1. P2L73: According to Table 1, the chemical shift δH 3.42 ppm was attributed to H-9' not H-9.

→ We corrected mentioned 1H NMR data at H-9 and H-9' positions.

  1. P2L84: The HMBC correlation of H-7' to C-2 was questionable. In my opinion, the correct one was H-7' to C-5. However, Figure S5 was too small for me to analyze it. Please check it.

→ We again confirmed the HMBC spectrum of compound 1. In our analyzed HMBC data, the correlation between H-7' to C-5 was shown. We added the expansion of HMBC spectrum in the supporting information.

  1. The citations [22] and [23] were missing before [24] in the text.

→ We corrected mentioned reference

  1. P2L85: The carbons C-4' and C-5' were missing when referred to ‘The HMBC correlations from the methylenedioxy moiety (H-10') indicate...’.

→ We added C-4', -5' data, and reference and the added word showed as red-color.

  1. P3: The relative configuration of C-8 was not determined, which should be done before the elucidation of its absolute configuration. Please check it.

→ In the case of the previously reported similar compounds, the relative configuration of arytetralin-type lignan is possible by comparison reference values along with 1H-1H coupling constants and NOESY correlations [1, 2].

  1. Latté, K.P.; Kaloga, M.; Schäfer, A.; Kolodziej, H. An ellagitannin, n-butyl gallate, two aryltetralin lignans, and an unprecedented diterpene ester from Pelargonium reniforme. Phytochemistry 2008, 69, 820-826
  2. Suh, W.S.; Kim, K.H.; Kim, H.K.; Choi, S.U.; Lee, K.R. Three new lignan derivatives from Lindera glauca (Siebold et Zucc.) Blume. Chim. Acta 2015, 98, 1087-1094.
  3. For compound 2, the interpretation of the methylene H2-9 in its 1H NMR spectrum was missing. And describe the key HMBC correlations in detail, not briefly mentioned as ‘Figure 2A’. The relative configuration of C-7' was not clearly determined.

→ We additionally described methylene protons of H-9 and key HMBC correlations. We added sentence showed as red-color. The relative configuration of compound 2 was also determined by comparing the NOESY correlations with reported references [3,4].

  1. Nhiem, N.X.; Lee, H.Y.; Kim, N.Y.; Park, S.J.; Kim, E.S.; Han, J.E.; Kim, S.H. Stereochemical assignment of five new lignan glycosides from Viscum album by NMR study combined with CD spectroscopy. Reson. Chem. 2012, 50, 772-777.
  2. Ullah, N.; Ahmad, S.; Anis, E.; Mohammad, P.; Rabnawaz, H.; Malik, A. A lignan from Daphne oleoides. Phytochemistry 1999, 50, 147-149.

  1. The citations [33] and [34] were missing before [35] in the text.

→ We corrected mentioned reference

Thank you for your kind and valuable comments to our manuscript. According to the valuable suggestions and comments, we have revised our manuscript. Changed words were expressed red ones. Please consider our following responses to each suggestion and comment.

Reviewer 3 Report

This is a very good manuscript. The research is very practical, the experiment is also well carried out, and the conclusion is well supported by the data. I suggest that the authors add previous reports on the inhibition of formation of AGEs and ONOO- in the manuscript, as well as related comparisons.

Author Response

Thank you for your kind and valuable comments to our manuscript. According to the valuable suggestions and comments, we have revised our manuscript
